# Distinct patterns of incidental exposure to and active selection of radicalizing information indicate varying levels of support for violent extremism

**Sandy Schumann** *, **Caitlin Clemmow, Bettina Rottweiler, Paul Gill**

University College London, London, United Kingdom

* s.schumann@ucl.ac.uk

## Abstract

Exposure to radicalizing information has been associated with support for violent extremism. It is, however, unclear whether specific information use behavior, namely, a distinct pattern of incidental exposure (IE) to and active selection (AS) of radicalizing content, indicates stronger violent extremist attitudes and radical action intentions. Drawing on a representative general population sample ($N$ = 1509) and applying latent class analysis, we addressed this gap in the literature. Results highlighted six types of information use behavior. The largest group of participants reported a near to zero probability of both IE to and AS of radicalizing material. Two groups of participants were characterized by high or moderate probabilities of incidental exposure as well as a low probability of active selection of radicalizing content. The remaining groups displayed either low, moderate, or high probabilities of both IE and AS. Importantly, we showed between-group differences regarding violent extremist attitudes and radical behavioral intentions. Individuals reporting near zero or high probabilities for both IE to and AS of radicalizing information expressed the lowest and strongest violent extremist attitudes and willingness to use violence respectively. Groups defined by even moderate probabilities of AS endorsed violent extremism more strongly than those for which the probability for incidental exposure was moderate or high but AS of radicalizing content was unlikely.

## Introduction

Radicalization refers to a process during which individuals adopt extremist attitudes and, in some but not all instances, "come to perceive acts of terrorism as a possible alternative for action" [1,2]. Exposure to messages that glorify or incite the use of violence, material that derogates or dehumanizes outgroup members, as well as exposure to ideological/terrorist propaganda is considered a risk factor for radicalization [3–5]. So-called exposure effects have been documented with respect to radicalizing content that individuals encountered online [6–10] and in prisons [11,12], or following interactions with peers and family members who endorse

**Funding:** This work was supported by the European Research Council under the European Union's Horizon 2020 research and innovation programme (Grant Agreement No 758834). The grant was awarded to PG. The funders had no role in study design, data collection and analysis, decision to publish, or preparation of the manuscript.

**Competing interests:** The authors report there are no competing interests to declare.

violent extremism and rule-breaking [13–16]. The present research builds on and advances those insights in two important ways.

Rather than explore the sites and actors through which radicalizing information is (made) available [17–19]), we focus on the processes by which individuals encounter such messages, that is, incidental exposure (IE) and active selection (AS) [4]. More precisely, considering a representative general population sample and relying on latent class analysis, we aim to identify different types (or classes) of information use behavior that are characterized by varying probabilities of IE to and AS of radicalizing material [20]. Additionally, we seek to examine which specific information use behavior is related with, and thus, indicative of, higher levels of violent extremist attitudes and a stronger willingness to use violence to attain collective goals.

## Effects of exposure to radicalizing information

Individuals are likely exposed to radicalizing information in settings where they can connect with radicalizing actors without interference (i.e., low formal or informal social control; [17]); it was shown that the latter include, for instance, prisons, schools, gyms, and coffee shops [11,12,18]. Given that hate and extremist actors have developed resilient cross-platform eco-systems online [21–24], the internet also provides numerous opportunities to be exposed to radicalizing messages [25–27]. Having said this, Clemmow and colleagues [28] showed that in a British sample of the general population, only 6.8% had, across their lifetime, sought violent extremist material online. A study of Belgian teenagers and young adults revealed that 96.2% had never actively pursued radicalizing material online; 49.5% had never been incidentally exposed [4,15].

Exposure to radicalizing information is considered a threat due to its expected impact on attitudes and behavior. Notably, interviews with individuals who had been radicalized highlighted that negative outgroup attitudes (here, rejection of non-Muslims or 'Western' politicians) as well as the willingness to use violence emerged in response to the consumption of terrorist propaganda [29]. Furthermore, cross-sectional survey studies demonstrated that actively seeking extremist content or engaging online in interactions with others who held extremist beliefs was associated with a higher likelihood of having had committed acts of political/religious violence against property or persons [9,15]. Exposure to right-wing extremist peers was further related to stronger right-wing extremist beliefs, which, in turn, predicted an increased likelihood of having had engaged in political violence [14]. Contrasting these findings, experimental work has failed to replicate (short-term) direct exposure effects [30].

In fact, a meta-analysis showed that exposure effects of radicalizing material are typically weak unless specific moderators are considered [31]. On the one hand, the impact of radicalizing information is stronger for those characterized by higher levels of thrill seeking [8] and trait aggression [31]. On the other hand, the frequency of exposure to radicalizing content likely shapes its influence. Specifically, research on the related problem of hate speech has found that repeated exposure to verbal aggressions predicts desensitization–negative emotions in response to such material are less likely [32]. Desensitization, in turn, was positively associated with prejudice towards the victims of hate speech [32]. That is, radicalizing information might, over time, lose its potential to shock and upset individuals. As the material becomes more palatable, it could elicit (stronger) attitude and behavior changes.

## Incidental exposure to and active selection of radicalizing information

In summary, exposure to radicalizing messages can facilitate, to some extent and for some individuals, the radicalization process. An underlying assumption of the aforementioned work

is that the process that enables individuals to be confronted with radicalizing information is universal. This, however, is not the case. Bouhana [17] provides initial guidance on this matter. More precisely, they stipulate that personal preferences or hobbies determine the likelihood of being present in particular physical or digital settings where radicalizing material may be disseminated (i.e., self selection effects [33]). Additionally, socio-demographic characteristics, such as one's educational level or age, or social group membership (e.g., ethnicity or religion), affect whether or not individuals engage in activities in certain places (i.e., social selection effects). Both self and social selection can enhance the *incidental exposure* to radicalizing messages. That is, individuals receive radicalizing information while they pursue other goals, such as their hobby or education [34].

Weeks and Lane [35] distinguish *state* and *trait unmotivated incidental exposure*. Exposure to radicalizing content is referred to as state unmotivated if individuals sympathize with or endorse (violent) extremist ideas but, in this very instance, seek to fulfil another need, such as entertainment (see [36], who refers to opportunistic information discovery that informs a 'background problem' while individuals aim to address a 'foreground problem'). Trait unmotivated incidental exposure implies that individuals are not at all sympathetic of violent extremism and encounter radicalizing messages when they seek different information or interaction partners. Following the *Political Incidental News Exposure Model* (PINE; [37]), effects of state and trait unmotivated exposure on attitudes and behavior are expected to vary.

More precisely, PINE [37] postulates that individuals appraise all information, including such that is encountered incidentally, to judge whether it is relevant, or "less important compared to the current processing goal" ([37]; p. 1039). Irrelevant information is disregarded and not processed any further (i.e., First level IE; [37]), although it might still be remembered especially after repeated exposure [38]. Relevant information is processed systematically (i.e., Second Level IE; [37]). In other words, sufficient cognitive resources are dedicated to paying attention, relating the new information to existing knowledge or memories, and scrutinizing the arguments (i.e., central route; [39]). Moreover, systematic processing may shift the original processing goal such that individuals actively seek further similar, radicalizing, information. Importantly, and in line with the elaboration-likelihood model, systematically processed information is more likely to shape lasting attitudes that are reliable predictors of long-term behavior [40].

In addition to being incidentally exposed, individuals also actively seek out content, sources, and media. Substantial evidence documents a general preference for information that aligns with existing beliefs and convictions (dissonance theory; [41]). One reason for the active selection of belief-congruent information is the defense of one's sense of self [42] and avoidance of threats to one's self-image (psychological immune system; [43]). The Meaning Maintenance Model [44] further stipulates that individuals "are inexhaustible meaning makers" (p. 91) and strive to keep previously established meaning frameworks intact; actively selecting information that confirms one's beliefs might serve as a strategy to achieve this.

Slater's [45,46] *Reinforcing Spiral Model* (RSM) provides a holistic framework to consider the consequences of belief-congruent information selection. First, seeking and consuming information in line with core attitudes is thought to raise the salience of the respective aspect of the self or identity. In turn, subsequent behavior, including future information selection, ought to be predicted by the salient beliefs–that is, belief-congruent information preferences should persist over time. Such ongoing selective information use predominantly contributes to the maintenance of existing attitudes at a stable level [47]. Attitudes might, however, change to more extreme positions if individuals experience a threat to their core beliefs or if they are embedded in information environments in which they hardly receive diverse viewpoints [45].

## The present research

Taken together, individuals may be exposed to radicalizing content as a result of its active selection as well as due to incidental exposure. Specifically, three types of information use behavior are conceivable [48,49] *(Hypothesis 1)*. First, the majority of the population is expected to never encounter radicalizing messages, neither through active selection nor incidental exposure [4,15]. Second, some individuals are thought to be only incidentally confronted with radicalizing information. Those individuals do not sympathize with the positions held by extremist or terrorist groups and, therefore, AS of radicalizing material is unlikely (trait unmotivated incidental exposure; [35]). Nonetheless, they might be connected with peer groups or frequent settings where radicalizing messages are disseminated (i.e., neutral magnets; [18]). A third groups of individuals is postulated to be exposed to radicalizing messages both through its AS and IE. This group agrees with the perspectives presented in radicalizing material, thus, AS of the information is likely as is state unmotivated incidental exposure in neutral or radicalizing magnets [18,35].

The three distinct information use behaviors–no AS of or IE to, only IE to, both AS of and IE to radicalizing messages–should relate to, or be indicators of, different levels of support for violent extremism *(Hypothesis 2a)*. Pauwels and Schils [15,50] showed, while controlling for other risk factors, significant positive associations between IE to as well as AS of extremist material online and reports of political violence against people or property. A higher frequency of active selection predicted a four and six-fold increase in the likelihood of violence against people and property respectively. A higher frequency of incidental exposure 'only' doubled the risk of political violence. The stronger relationship between active selection of extremist material and political violence is not surprising; AS of such content suggests that the individuals already endorsed extremist opinions. Consequently, information use behavior that is characterized at least to some degree by the active selection of radicalizing information is expected to be associated with stronger support for violent extremism than patterns defined by no or only incidental exposure *(Hypothesis 2b)*.

## Method

To test these hypotheses, we conducted an online survey. The analyses described below were not pre-registered. The analytical code, material, and data as well as all supplementary material are available here: https://osf.io/jxsva/?view_only=bb8209691cd44cefb64da484aa93a05f

Ethics approval was granted by the UCL, Department of Security and Crime Science ethics committee (approval number: n/a). Participants gave written informed consent.

## Sample

A total of $N = 1509$ participants, reflecting the U.K. general population in terms of age, gender, and ethnicity distribution, were recruited. Representativeness is based on the most recent census data at the time of data collection, 2011. The survey encompassed eight attention checks; participants who failed three or more attention checks were excluded. The final analytical sample included $N = 1495$ participants. Overall, 51.4% participants identified as female, 47.6% identified as male, 0.3% indicated non-binary/third gender as their gender status, 0.1% self-described their gender status, and 0.3% preferred not to answer the question. The sample was on average $M_{age} = 45.04$ ($SD_{age} = 15.65$, range: 18–78) years old. The majority of participants (80.4%) stated 'White' as their ethnicity. This was followed by 6.9% who reported 'Asian', and 3.4% participants who identified as 'Black'. Further demographic information is available in the supplementary material (S1).

## Measures

To examine the frequency of *active selection* of and *incidental exposure* to radicalizing material we relied on an adapted version of the EXPO-12 scale [4]. Frequency of AS of radicalizing information was measured with 14 items (e.g., 'Searched for books, magazines, or other types of text which support the use of violence to achieve political, religious, or social goals', 'Searched for content online like websites, memes, or videos that support the use of violence to achieve political, religious, or social goals', 'Searched for podcasts, songs, or other types of audios which support the use of violence to achieve political, religious, or social goals', 'Searched for content made by people who have committed violence to achieve political, religious, or social goals, such as manifestos, or YouTube videos'; *1* = Never, *2* = Once or twice, *3* = A few times, *4* = Once or twice a year, *5* = Once or twice a month, *6* = Once or twice a week, *7* = Every day). We assessed frequency of incidental exposure to radicalizing information with nine items (e.g., 'Received printed texts which you didn't ask for such as books or magazines which support violence to achieve political, religious, or social goals', 'Received content online that you didn't ask for such as images or videos which show acts of violence to achieve political, religious, or social goals', 'Overheard people expressing views in support of violence to achieve political, religious, or social goals', 'Accidentally witnessed comments being made online to support violence to achieve political, religious, or social goals'; *1* = Never, *2* = Once or twice, *3* = A few times, *4* = Once or twice a year, *5* = Once or twice a month, *6* = Once or twice a week, *7* = Every day). For the AS and IE measures, five items respectively referred specifically to online settings.

*Support for violent extremism* was captured with two outcome variables. Participants, first, completed the four items of the radicalism dimension of the activism and radicalism intention scales (ARIS; [51]), reporting the extent to which they agreed with statements such as 'I would participate in a public protest against oppression of my group even if I thought the protest might turn violent' or 'I would continue to support an organisation that fights for my group's political and legal rights even if the organisation sometimes resorts to violence' (*1* = strongly disagree, *7* = strongly agree; α = .86). We further introduced the violent extremist attitude scale [52], a four-item measure that explores attitudes towards the use of violence to pursue political, religious, or social justice goals (e.g., 'It's sometimes necessary to use violence to fight against things that are very unjust', 'It's OK to support groups that use violence to fight injustice'; *1* = strongly disagree, *7* = strongly agree; α = .87).

Three other measures for violent extremism were included in the survey. Result patterns largely replicated the ones demonstrated for the two aforementioned scales; please see supplementary material S2 for details and discrepancies. The complete survey materials are presented in the supplementary material (S3).

## Procedure

Data were collected in September 2021 on Prolific Academic, an online opt-in access panel [53]. We choose Prolific as a platform for recruiting participants because it has been consistently found to provide high(er)-quality data, especially as compared to other providers such as Amazon Mechanical Turk and CloudResearch [54,55]. Participants filled in the survey in their own time, without supervision. Participants could not skip questions or items. Questions regarding active information selection and incidental exposure were asked before the dependent variables. Demographic information was assessed in short blocks throughout the survey to break up the more demanding questions on information behavior and support for violent extremism. The average completion time was 14 minutes.

## Results

### Analytical approach

We employed a latent class analysis (LCA) to identify unobserved sub-groups (i.e., latent classes) of individuals based on observable characteristics [56] here, the reported frequency of active selection of as well as incidental exposure to radicalizing information. Specifically, it is expected that individuals' scores on the observable variables are an indicator for their membership in a particular latent class [57]. The number of classes that are to be attained in a LCA is not pre-defined. Instead, several solutions with a different number of classes are estimated to then determine which model fits the observed data best. To estimate the models, we used maximum likelihood estimation with robust standard errors. Evaluations of the fit of different solutions were based on two criteria, the sample size adjusted Bayesian information criterion (aBIC) and consistent Akaike information criterion (cAIC; [58]). A lower aBIC and cAIC indicated a better model fit. Lastly, entropy was assessed to understand the extent to which, or the uncertainty with which, a chosen solution accurately defined the different latent classes. A threshold of .80 suggested acceptable entropy [59]. It should be noted that individual fit statistics often suggest different optimal class solutions. Hence, fit statistics were considered alongside knowledge from previous research and theory to identify the most appropriate class solution. After identifying the optimal number of classes, participants were assigned to one specific class, based on the highest class-membership probability. Following, we conducted a multi-variate analysis of variance to investigate between-group differences on the two dependent variables.

Latent class analysis relies on categorical variables. As described previously, the survey measures of AS and IE offered Likert-type answer options. However, the distribution of the variables indicated that they were strongly right-skewed; the large majority of participants indicated that they had 'Never' engaged with or were 'Never' confronted with radicalizing information (Table 1). We therefore decided to recode the variables into dichotomous measures that indicated whether participants had responded 'Ever' (i.e., answer options *2* to *6*) or 'Never' (i.e., answer option *1*) [60].

All analyses were completed with *R* 4.1.0 (R Core Team, 2021). Relevant packages are indicated in the analytical script. The dataset used for the analysis reported below included no missing values.

### Descriptive statistics

Table 1 documents the means and standard deviations of the outcome variables as well as the prevalence of different indicators of active selection of and incidental exposure to radicalizing information in the whole sample. Results showed overall low support for violent extremism. Moreover, active selection of radicalizing content was relatively uncommon whilst incidental exposure appeared to be more frequent. Additionally, there was noticeable variation between certain types of activities. For instance, active selection of information about individuals who had committed acts of violence was 12 times more common than searching for information about weapons.

### Unique patterns of AS of and IE to radicalizing information

We estimated models for up to eight latent classes of information use behavior. All models attained entropy values above the threshold of .80 [59]. The aBIC and CAIC were on balance the lowest for the six classes solution (Table 2), that is, the sample was distinguished into six groups, each characterized by unique information use behavior.

**Table 1.  Descriptive statistics of information use behavior and dependent variables.**

| | Mean | SD |
|---|---|---|
| **Radicalism Intention Scale** | 2.18 | 1.23 |
| **Violent Extremist Attitude Scale** | 2.74 | 1.31 |
| **Active Selection** | **% (*n*) Never** | **% (*n*) Ever** |
| 1) Searched for books, magazines, or other types of text which support the use of violence to achieve political, religious, or social goals | 88.7 (1326) | 11.3 (169) |
| 2) Had discussions in person with people who support the use of violence to achieve political, religious, or social goals | 72.0 (1077) | 28.0 (418) |
| 3) Used the internet to chat online with people who support the use of violence to achieve political, religious, or social goals | 92.4 (1381) | 7.6 (114) |
| 4) Searched for content online like websites, memes, or videos that support the use of violence to achieve political, religious, or social goals | 85.8 (1282) | 14.2 (213) |
| 5) Searched for places where people who support the use of violence to achieve political, religious, or social goals spend time | 93.0 (1391) | 7.0 (104) |
| 6) Searched online for groups or people who support the use of violence to achieve political, religious, or social goals | 88.6 (1324) | 11.4 (171) |
| 7) Used the internet to observe online chat between other people who support the use of violence to achieve political, religious, or social goals | 82.5 (1234) | 17.5 (261) |
| 8) Searched for images or videos of violence to achieve political, religious, or social goals | 83.9 (1254) | 16.1 (241) |
| 9) Searched for podcasts, songs, or other types of audios which support the use of violence to achieve political, religious, or social goals | 90.4 (1351) | 9.6 (144) |
| 10) Searched offline (in real life) for people or groups who support the use of violence to achieve political, religious, or social goals | 97.2 (1453) | 2.8 (42) |
| 11) Searched for information on how to use weapons or make bombs for violence to achieve political, religious, or social goals | 97.9 (1463) | 2.1 (32) |
| 12) Searched for content made by people who have committed violence to achieve political, religious, or social goals, such as manifestos, or YouTube videos | 86.4 (1292) | 13.6 (203) |
| 13) Searched for content about people who have committed violence to achieve political, religious, or social goals | 75.7 (1131) | 24.3 (364) |
| 14) Searched for events or activities to attend which support violence to achieve political, religious, or social goals | 96.3 (1440) | 3.7 (55) |
| **Incidental Exposure** | **% (*n*) Never** | **% (*n*) Ever** |
| 1) Received printed texts which you didn't ask for such as books or magazines which support violence to achieve political, religious, or social goals | 87.5 (1308) | 12.5 (187) |
| 2) Accidentally come across content which supports violence to achieve political, religious, or social goals online | 60.8 (909) | 39.2 (586) |
| 3) Received content online that you didn't ask for, such as memes or videos which violence to achieve political, religious, or social goals | 70.0 (1046) | 30.0 (449) |
| 4) Had content which supports violence to achieve political, religious, or social goals recommended to you on social media | 82.6 (1235) | 17.4 (260) |
| 5) Received content online that you didn't ask for such as images or videos which show acts of violence to achieve political, religious, or social goals | 73.3 (1096) | 26.7 (399) |
| 6) Received audio content that you didn't ask for such as podcasts or songs which support violence to achieve political, religious, or social goals | 90.6 (1354) | 9.4 (141) |
| 7) Come across content online about using violence to achieve political, religious, or social goals while looking for content about something else | 69.0 (1032) | 31.0 (463) |
| 8) Overheard people expressing views in support of violence to achieve political, religious, or social goals | 61.6 (921) | 38.4 (574) |

(*Continued*)

**Table 1.** (Continued)

| | Mean | SD |
|---|---|---|
| 9) Accidentally witnessed comments being made online to support violence to achieve political, religious, or social goals | 62.1 (928) | 37.9 (567) |

To interpret the concrete patterns of IE to and AS of radicalizing information for the six groups, we inspected the conditional item response probabilities (Fig 1). Detailed numeric values are available in supplementary material (S4). Accordingly, the six classes are described as follows: (Class 1) a near zero probability of either AS or IE, (Class 2) a moderate probability of both AS and IE, (Class 3) a moderate probability of AS and a high probability of IE, (Class 4) a low probability of AS and a moderate probability of IE, (Class 5) a high probability of both AS and IE, and (Class 6) a moderate probability of AS and a high probability of IE. Class 1 is the largest, Class 5 the smallest (Fig 1). Importantly, there is evidence in support of Hypothesis 1. The six classes reflect the proposed three information use patterns, specifically, neither AS nor IE (Class 1), IE only (Classes 3 & 4), or a combination of active selection of and incidental exposure to radicalizing information (Classes 2, 5, & 6) (Fig 1).

### AS, IE, and support for violent extremism

Next, we examined between-group differences with respect to support for violent extremism (*Hypothesis 2a, 2b*). The multi-variate analysis of variance demonstrated main effects for the radicalism intentions scale ($F(5, 1489) = 69.04$, $p < .001$, $\eta^2 = .188$) and the violent extremist attitudes scale ($F(5, 1489) = 41.76$, $p < .001$, $\eta^2 = .123$) (see Table 3 for the mean scores in each class). Post-hoc tests (Table 4) further showed that participants in Class 1 reported overall the lowest and those in Class 5 the highest support for violent extremism. Indeed, all between-group differences were statistically significant with the following exceptions: we identified no differences between Class 2 and 6, and between Class 3 and 4 for both outcome variables, as well as no difference between Class 2 and 3 for the dependent variable 'attitudes towards violent extremism'. The highly similar result pattern for both outcome variables is not surprising; indeed, an exploratory factor analysis with oblimin rotation showed that all items measuring the two dependent measures loaded on one factor (see Supplementary Material S5).

### Discussion

The present study examined patterns of incidental exposure to and active selection of radicalizing messages in the general population. We showed that individuals exhibit three broad

**Table 2. Evaluating class solutions.**

| Model | LL | BIC | aBIC | cAIC | likelihood-ratio | entropy |
|---|---|---|---|---|---|---|
| 1 Class | -14585.72 | 29339.57 | 29266.50 | 29326.57 | - | - |
| 2 Classes | -11227.60 | 22798.76 | 22649.46 | 22845.76 | 8585.29 | .92 |
| 3 Classes | -10480.18 | 21479.36 | 21253.82 | 21550.36 | 7090.46 | .91 |
| 4 Classes | -10218.03 | 21130.49 | 20828.71 | 21225.49 | 6566.15 | .90 |
| 5 Classes | -10055.06 | 20980.00 | 20601.97 | 21099.00 | 6240.22 | .88 |
| **6 Classes** | -9943.47 | 20932.26 | 20477.99 | 21075.26 | 6017.05 | .88 |
| 7 Classes | -9878.84 | 20978.42 | 20477.91 | 21145.42 | 5887.77 | .88 |
| 8 Classes | -9815.15 | 21026.49 | 20419.74 | 21217.49 | 5760.40 | .87 |

*Note*. LL = log-likelihood, aBIC = sample-size adjusted BIC, cAIC = consistent Akaike information criterion; highlighted in bold is the model with the best overall fit.

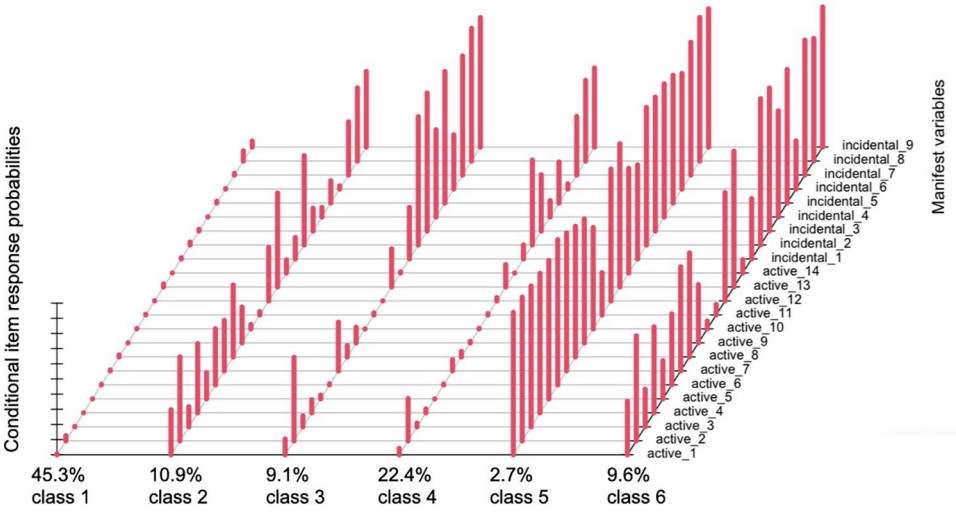

**Fig 1. Conditional item response probabilities.**

information use behaviors, namely, a near zero probability of exposure, varying probabilities of only IE to, or both AS of and IE to radicalizing information. As demonstrated in previous research, most participants indicated neither active selection of nor incidental exposure to radicalizing content [4,15]. Importantly, and in line with our hypothesis, information use characterized by even low probabilities of active selection of radicalizing material was associated with stronger violent extremist attitudes and a higher willingness to use violence to attain collective goals than patterns defined by no exposure or only incidental exposure to radicalizing information.

These findings have several implications. Notably, we identified no group of participants that reported near zero probability of IE but at least a moderate probability of active selection of radicalizing content. Thus, our results suggest that individuals who are actively seeking radicalizing information also pursue other information goals, such as entertainment or social interaction, in settings where radicalizing content is available—state unmotivated incidental exposure is as likely as AS [35]. Evidence of moderate or high probabilities of IE to *and* AS of radicalizing information, therefore, could be indicative of overall changes in individuals' habits and social life, "a slow marginalization away from conventional society and toward a much narrower society where extremism becomes all-encompassing" ([61], p. 89, [62,63]).

Table 3. Means and standard deviation of support for violent extremism per class.

| Class | Radicalism Intentions Scale *M* (*SD*) | Attitudes towards Violent Extremism *M* (*SD*) |
|---|---|---|
| 1 | 1.76 (.96) | 2.35 (1.21) |
| 2 | 2.84 (1.38) | 3.32 (1.36) |
| 3 | 2.38 (1.22) | 2.95 (1.18) |
| 4 | 2.13 (1.07) | 2.73 (1.21) |
| 5 | 4.20 (1.54) | 4.31 (1.44) |
| 6 | 2.87 (1.26) | 3.37 (1.21) |

*Note*. Class 1—no probability of either AS or IE, Class 2—moderate probability of both AS and IE, Class 3—moderate probability of AS, high probability of IE, Class 4—low probability of AS, moderate probability of IE, Class 5—high probability of both AS and IE, Class 6—moderate probability of AS, high probability of IE

**Table 4. Post-hoc between-group comparisons.**

| Reference class | Comparison class | Radicalism Intentions Scale Mean difference; 95% CI; *p* | Attitudes towards violent extremism Mean difference; 95% CI; *p* |
|---|---|---|---|
| 1 | 2 | -1.08; -1.42, -.76; < .001 | -.98; -.132, -.63; < .001 |
|  | 3 | -.62; -.94, -.30; < .001 | -.60; -.92, -.28; < .001 |
|  | 4 | -.38; -.57, -.18; < .001 | -.38; -.61, -.15; < .001 |
|  | 5 | -2.44; -3.19, -1.70; < .001 | -1.96; -2.66, -1.25; < .001 |
|  | 6 | -1.12; -1.44, -.79; < .001 | -1.02; -1.34, -.70; < .001 |
| 2 | 3 | .46; .02, .90; .034 | .37; -.05, .80; .126 |
|  | 4 | .71; .35, 1.07; < .001 | .60; .23, .96; < .001 |
|  | 5 | -1.36; -2.16, -.56; < .001 | -.98; -1.73, -.23; .004 |
|  | 6 | -.04; -.47, .40; 1.00 | -.04; -.47, .37; 1.00 |
| 3 | 4 | .25; -.10, .59; .315 | .22; -.13, .57; .450 |
|  | 5 | -1.82; -2.61, -1.03; < .001 | -1.36; -2.10, -.61; < .001 |
|  | 6 | -.49; -.92, -.07; .013 | -.42; -.83, -.004; .046 |
| 4 | 5 | -2.07; -2.82, -2.32; < .001 | -1.58; -2.29, -.86; < .001 |
|  | 6 | -.74; -1.09, -.39; < .001 | -.64; -.98, -.29; < .001 |
| 5 | 6 | 1.33; .53, 2.12; < .001 | .94; .20, 1.69; .006 |

Additionally, we observed that a higher probability of either IE or AS per se was not associated with corresponding stronger support for violent extremism. For instance, the classes of participants characterized by a moderate and high probability of incidental exposure to radicalizing messages did not express different levels of violent extremist attitudes or radical action tendencies. Instead, significant differences were found between the group that noted high probabilities for both IE and AS and all others, as well as between groups that did and did not report active selection of radicalizing material. Hence, even a low probability of active selection of radicalizing information likely reflects noteworthy commitment to the use of violence.

Although we demonstrated the three proposed patterns of information use behavior (i.e., neither IE or AS, only IE, AS and IE), the sample was distinguished into six groups, two of which were defined by different probabilities of only IE and three of which were characterized by low, moderate, or higher probabilities of both actively seeking and being incidentally exposed to radicalizing material. First, it is important to not mistake this result as differences in the *frequency* of incidental exposure or active selection. Second, it is worth considering the factors that contribute to the emergence of low, moderate, or high probabilities of certain information use patterns. With regards to incidental exposure to radicalizing information, self and social selection processes, that is, personal preferences as well as characteristics pertaining to social group memberships, determine the likelihood of being present in places where radicalizing agents disseminate their messages, even if the person is not interested in the content [17]. Unfortunately, previous research on the topic of violent extremism has considered 'exposure' primarily as a predictor and not a dependent variable. Therefore, the exact correlates of IE have not been examined. Clemmow and colleagues' [5] study is an exception that should be extended in future research. They demonstrated that younger individuals were more likely to be incidentally exposed or actively select radicalizing messages and that male respondents were more likely to be exposed than female participants.

Continuing the previous line of thought, the availability (or lack thereof) of alternative settings to fulfill personal interests, practice one's religion, or receive an education should also affect the probability of repeated IE to radicalizing information. That is, a higher probability of incidental exposure to radicalizing material in absence of AS could indicate that despite finding the content irrelevant, individuals are not changing (or are not able to change) their information environments. The latter might be the result of close peers or family members endorsing extremism, which constitutes a significant risk factor of radicalization [14–16,64].

Further, following the reinforcing spirals model [45,46], different probabilities of AS of radicalizing information could suggest varying levels of salience of beliefs or aspects of one's identity that are associated with a violent extremist ideology. It is expected that selective information use is associated with the maintenance of violence-affirming or violent extremist attitudes at a stable level over time [47]. However, the probability of AS should increase when information environments become more homogenous, devoid of diverse views, which also predicts a shift of attitudes to more extreme positions (i.e., positive feedback loop; [46]). Hence, as mentioned earlier, a higher probability of active selection of radicalizing materials could imply that individuals are more embedded in networks or communities where violent extremism is endorsed. Future research should aim to investigate this speculation empirically and, more generally, consider the antecedents, rather than only the consequences, of incidental exposure to as well as active selection of radicalizing material.

Any practical implications that are based on one study or analysis can only be preliminary. We nevertheless believe that the results inform two fields of practice. First, we contribute insights to the development of individual risk assessment tools that aim to identify those vulnerable to violent extremism before they commit violence [65]. These tools (e.g., VERA(2) [66]; TRAP-18 [67]) typically include a wide variety of risk or protective factors, such as grievances and ideological commitment, criminal history, mental health, or the presence of a social support system. Previous research has further developed approaches to assess individuals' risk based on textual digital trace data (i.e., linguistic risk assessment; [68,69]). Our findings suggest that information use behavior, which is to some extent observable, should be considered as well, and in a more nuanced way, in individual risk assessment and management. Namely, we showed that evidence of patterns of only incidental exposure to radicalizing material might be less of a concern, especially if it remains temporary (i.e., individuals are not likely to be embedded in environments where radicalizing messages are regularly disseminated). In turn, information use behavior defined by even a low probability of active selection of radicalizing content online or offline likely indicates stronger support for violent extremism which could manifest in acts of terrorism. Here, efforts to prevent radicalization are especially pertinent.

Perhaps somewhat ironically, in online settings, the fact that individuals actively seek radicalizing information could provide an avenue for targeted prevention. Notably, redirect(ion) projects, which have been implemented in collaboration with large social media platforms, use the search of keywords that are thought to be indicative of (violent) extremist or terrorist actors and narratives as a cue to provide counter-narratives or offer support by civil society organizations specialized in disengagement [70]. Evaluations of past programs highlight that their implementation is not without challenges [71,72]. However, our results indicate that using the active search for radicalizing content as one of several indicators of risk is highly promising.

The aforementioned conclusions must of course be considered in light of some methodological limitations. Self-report data can be affected by a range of biases, including concerns of social desirability (i.e., under-reporting of exposure to radicalizing information and support for violent extremism), inability to accurately estimate the frequency of past information use, as well as fatigue. We addressed the latter by including several attention checks, which only very few participants failed. Nonetheless, we must acknowledge that the exact size of the different groups might not be estimated accurately as participants did not report (all) information use behavior correctly. Similarly, underreporting of support for violent extremism could mean that between-group differences are underestimated. Future research might therefore consider daily diary studies to avoid recollection biases of information use behavior.

Further, the study design only permits correlative conclusions. We do not suggest that IE to or AS of radicalizing information predict (or even cause) support for violent extremism.

However, as elaborated in detail in the previous sections, the different observable information use patterns–for instance, evidence of even a moderate probability of active selection of radicalizing content–could be viewed as indicators of individuals' (higher) level of radicalization [73]. In order to understand whether active selection of radicalizing material is not merely indicative of but causes stronger support for violent extremism, longitudinal observational or experimental studies are required.

Additionally, we did not focus on, or distinguish between, particular ideologies or exposure to information from specific extremist and terrorist actors. Although we would not expect differences in information use patterns, it is conceivable that especially state unmotivated incidental exposure is more prevalent for ideologies that are embedded in the mainstream public discourse. In the U.K. context, where our data was collected, this might apply especially to white supremacist or anti-immigrant messages. Going forward, the patterns that we have identified should be replicated across specific types of radicalizing content and ideologies. Relatedly, individuals' information use behavior is shaped to some extent by the cultural, political, and economic context, for instance, by governmental policies regarding what information is available or permissible in certain settings. Thus, the distinct classes, defined by varying probabilities of IE and AS, are perhaps only representative of information use behavior in a Western democracy. We, therefore, encourage others to replicate the study in other geographical and political contexts.

Lastly, it is recommended to explore the research questions with other samples, notably, populations that have adopted (violent) extremist beliefs more strongly than the general population (e.g., individuals who are part of a de-radicalization program) or individuals who have committed or prepared acts of terrorism. In those instances, we would expect the absence of information use patterns that are defined by a near zero probability of incidental exposure to and active selection of radicalizing information or only evidence of incidental exposure.

## Conclusion

Our research showed that a substantial proportion of the population is never exposed to information that incites or condones the use of violence to attain collective goals. We further demonstrated that not all individuals who encounter such material are per se at risk of radicalization. More precisely, information use behavior that is characterized by the active selection of (rather than only incidental exposure to) radicalizing content was associated with significantly higher support for violent extremist attitudes and radical behavior intentions. Efforts to prevent and counter radicalization should, therefore, focus in particular on individual for whom it is evident that they actively seek radicalizing material.

## Author Contributions

**Conceptualization:** Sandy Schumann, Caitlin Clemmow, Bettina Rottweiler, Paul Gill.

**Data curation:** Bettina Rottweiler.

**Formal analysis:** Caitlin Clemmow.

**Funding acquisition:** Paul Gill.

**Methodology:** Caitlin Clemmow.

**Project administration:** Sandy Schumann.

**Visualization:** Sandy Schumann.

**Writing – original draft:** Sandy Schumann, Bettina Rottweiler.

**Writing – review & editing:** Sandy Schumann, Caitlin Clemmow, Bettina Rottweiler, Paul Gill.

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
