## [Decision Letter · Decision Letter 0]

1 Sep 2023

PONE-D-23-12940Distinct patterns of incidental exposure to and active selection of radicalizing information indicate varying levels of support for violent extremismPLOS ONE

Dear Dr. Schumann,

Thank you for submitting your manuscript to PLOS ONE. I sincerely apologise for the unusually delayed review timeframe. Your manuscript has been assessed by four reviewers, whose comments are appended below. The reviewers comment positively on the approach and the perceived interest in the study topic, and are generally positive about the overall strength of the conclusions. However, they raise a number of concerns that should be addressed. Therefore, we invite you to submit a revised version of the manuscript that addresses the points raised during the review process.

We look forward to receiving your revised manuscript.

Kind regards,

Emily Chenette

Editor in Chief

PLOS ONE

Journal Requirements:

Reviewers' comments:

Reviewer's Responses to Questions

**Comments to the Author**

1. Is the manuscript technically sound, and do the data support the conclusions?

Reviewer #1: Yes

Reviewer #2: Yes

Reviewer #3: Yes

Reviewer #4: Partly

2. Has the statistical analysis been performed appropriately and rigorously? 

Reviewer #1: Yes

Reviewer #2: Yes

Reviewer #3: I Don't Know

Reviewer #4: Yes

3. Have the authors made all data underlying the findings in their manuscript fully available?

Reviewer #1: Yes

Reviewer #2: Yes

Reviewer #3: Yes

Reviewer #4: Yes

4. Is the manuscript presented in an intelligible fashion and written in standard English?

Reviewer #1: Yes

Reviewer #2: Yes

Reviewer #3: Yes

Reviewer #4: Yes

5. Review Comments to the Author

Reviewer #1: This research study is noteworthy for its effective demonstration of the probabilities associated with an individual's incidental exposure to violent extremism (VE) and their active selection of radicalizing content. However, in terms of extremism, the study lacks descriptive data on the sample's profile within a population, which would be valuable in determining when evidence of exposure could indicate an increased risk of violence. Notably, the study findings suggest that individuals who only experience incidental exposure to radicalizing material are less likely to raise concerns. To strengthen the identified correlations, it is crucial to provide a comprehensive elucidation of the sample profile, which consists of 1,509 respondents. Given the significance of radicalism and violent extremism within religion-based or race-political-based ideologies, an examination of the sample characteristics becomes imperative.

Reviewer #2: I congratulate the authors (and thank them, because this was an easy, interesting, and enjoyable read!) for putting together a coherent and well-written paper on a useful and novel topic. This is a high-quality paper and I am recommending minor revisions, which I list below.

My only substantial constructive criticism is that the discussion section covers theoretical considerations and implications fairly thoroughly, but does not touch on practical implications that the results of this research may have for PCVE interventions and policies to reduce the effects of exposure to VE content. In my opinion, the crux of the paper is summarized in the following sentence (end of page 21):

“Hence, even a low probability of active selection of radicalizing information is thought to reflect noteworthy commitment to the use of violence.”

I would like to see some additions to the Discussion (even if only a paragraph or two) thoughtfully addressing these results in the context of UK PCVE efforts, in particular when considering the broad-reaching, public health style approach to steering people away from environments and individuals that provide radicalizing information. Some examples off the top of my head could include programs such as those that promote healthy and safe internet usage among school-age children or ad campaigns (as I have seen on the London Underground for a variety of social ills, including hate crimes) reminding people of the potential legal penalties for seeking out VE content.

“In addition to being incidentally exposed, individuals also actively select content, 147 sources, and media. In doing so, they aim to gratify situational needs or motives informed by 148 personality traits, interpersonal relationships, their social identities, societal values, norms, or 149 laws (i.e., uses and gratification theories) [39-41].”

In the above section, I don’t find uses and gratification theories very compelling as an explanatory theory, but I do find significance quest or more generally the need for meaning, which align very closely with dissonance theory and RSM, to be a far better fit. From where I sit, a motivational (i.e., psychological motivation) approach makes good sense there. In fact, several motivational models having to do with meaning (Psychological Immune System, see Gilbert et al., 1998; Porot & Mandelbaum, 2021; Meaning Maintenance Model, see Heine, Proulx, & Vohs, 2006; and Meaning-Making Mode, see CL Park, 2010) may be a better fit for explaining the drive for belief congruence a la dissonance theory/confirmation bias. That is, people are motivated to select and pursue information that supports their beliefs due to our inherent need for meaning (i.e., congruence, closure, purpose, and significance).

The following criticism is more of a procedural one, and not much to do with the paper itself. You mention that,

“249 Three other measures for violent extremism were included in the survey. Result 250 patterns largely

replicated the ones demonstrated for the two aforementioned scales; see 251 supplementary material S2 for

details and discrepancies. The complete survey materials are 252 presented in the supplementary material S3. We recorded no information that would allow us 253 to identify participants at any stage of the project.”

As a fellow VE researcher, I wanted to take a look at the other VE scales and observe the result patterns you mentioned, but was taken aback to see that S2 didn’t contain the results of statistical analyses (i.e., tables), but rather raw data and a codebook (unless I missed something). This was disappointing because I wanted to check out those findings for myself, but was not able to quickly and easily do so (I’m sure you can relate to the fact that I didn’t want to have to download the data, create the necessary variables from the individual items, and run the analyses myself). I think, if you’re going to say the results basically replicated across the multiple VE measures then you should provide those results, or simply not mention that. But that may just be my opinion, so feel free to take that with a grain of salt!

I’m not familiar with Prolific and would like you to add a sentence or two describing its data quality and reliability compared to other platforms (as you would for Qualtrics, MTurk, etc.).

“Data were collected in September 2021 on Prolific Academic, an online opt-in access 256 panel.”

On page 21, I think the following is supposed to read “entertainment or social interaction”:

“However, our results suggest that individuals who are actively seeking radicalizing information also pursue other information goals, such as entertainment of social interaction…”

Reviewer #3: This paper is extremely timely as it explores the processes by which individuals encounter radicalising contents online, i.e. incidental exposure (IE) and active selection (AS). The authors have developed and adopted a suitable methodology that allowed them to study a wide sample of population (N=1509) and build six types of information use behaviour, reflecting different levels of IE and AS.

The authors hypotheses, methodological approach and claims were all sufficiently contextualised in the relevant literature, which was reviewed and used to build their own model of analysis. The original data were deposited appropriately and access to the repository was provided.

The findings presented some relevant insights into the categories explored. This research revealed a strong relationship between AS and political violence as people with higher AS already hold opinions that are aligned with those presented in radicalizing content. Also, this research showed that individuals who are actively seeking radicalizing information also pursue other information goals, such as entertainment of social interaction, in settings where radicalizing content is available.

With regard to IE to radicalising information, this research showed that personal preferences and membership to specific groups determine the likelihood of being present in places where radicalizing agents disseminate their messages, even if the person is not interested in the content.

While as a qualitative researcher I do not feel I have got extensive quantitative knowledge to comment on the rigour of the statistical analysis performed, I can certainly say that the manuscript is well organised, clearly written, understandable and accessible even to non-specialist audience.

The method utilised in this paper is replicable and the authors themselves have invited other scholars to use their methodology and replicate the study in different geographical and political contexts to enhance the generalizability of their findings.

A few considerations/points below could help strengthen this paper further:

1. On p.2, line 33 it is stated “Exposure to radicalizing information has been associated with support for violent extremism”, however the authors have not provided any reference for this statement. Who says so? No references were provided. Is that their assumption or substantiated by previous studies? There are numerous studies exploring online ecosystems that suggest a link between radicalisation contents and direct action, however this is just one among the numerous triggers to violence (See: See Kenyon, Jonathan, Jens Binder, and Christopher Baker-Beall. 2022. "Understanding the Role of the Internet in the Process of Radicalisation: An Analysis of Convicted Extremists in England and Wales." Studies in Conflict & Terrorism:1-25. doi: 10.1080/1057610X.2022.2065902; Williams, Thomas James Vaughan, Calli Tzani, Helen Gavin, and Maria Ioannou. 2023. "Policy vs reality: comparing the policies of social media sites and users’ experiences, in the context of exposure to extremist content." Behavioral Sciences of Terrorism and Political Aggression:1-18. doi: 10.1080/19434472.2023.2195466.)

2. P.3, lines 57-58 state: “Radicalization refers to a process during which individuals develop a growing willingness to use violence as a means to pursue political, religious, or social-justice goals”. This definition seems to assume that radicalisation is a natural process towards violent actions and terrorism, which is not often times. The wide cohort of non-violent but extreme individuals and groups are an example of what stated above and implies that radicalisation should not be considered as a natural path towards violence. (See: Orofino, Elisa, and William Allchorn. 2023. Routledge Handbook of Non-violent Extremism: Groups, Perspectives and New Debates: Taylor & Francis; Littler, Mark, and Ben Lee. 2023. "Studying Extremism in the 21st Century: The Past, a Path, & Some Proposals." Studies in Conflict & Terrorism:1-3. doi: 10.1080/1057610X.2023.2195056.)

Some other comments/considerations include the following:

- On p. 28 we read “This study did not focus on, or distinguish between, particular ideologies or exposure to information from specific extremist and terrorist actors. Although we would not expect differences in information use patterns, it is conceivable that especially state unmotivated incidental exposure is more prevalent for ideologies that are embedded in the mainstream public discourse”. It would be good to test the results of this study using specific ideologies (e.g. Islamism, far-right, eco-radicalism) sharing common elements (enemy to fight, victims to protect and call urgent action) but coming from very different ideological tenets (See Orofino, Elisa. 2022. "Extremism (s) and Their Fight against Modernity: The Case of Islamists and Eco-Radicals." Religions 13 (8).)

- This study was conducted on general population and it served the purpose it was conceived for. However, it would be useful if future research could survey Prevent population (Channel adopted cases in the UK) to verify level of AS and IE to radicalising contents in the pathways towards violent actions (not terrorism) and extreme behaviours. Channel adopted cases are made up of people who have been referred to Counter-terrorism policing because holding extreme ideas but who have not committed any offence (yet). In the UK, these people are supported by an early de-radicalisation programme (Channel) tailored on their specific case.

-Finally, this research showed that individuals with higher AS of radicalising content also pursue other information goals, such as entertainment of social interaction, in settings where radicalizing content is available. This finding suggests that these people act as a vehicle to spread radicalising contents to others in all the platforms they navigate. As a result, more research on specific online ecosystems (e.g. bodybuilding websites, gaming platforms) is needed.

Reviewer #4: Thank you for the opportunity to review this article that explores the distinct patterns of incidental exposure (IE) and active selection (AS) of radicalizing content and their impact on radicalization outcomes. The article demonstrates a strong understanding of the relevant literature and presents an intriguing setting for investigation.

One important methodological concern I have with the article relates to the similarity between the items measuring the active selection and support for violent extremism. The potential overlap in meaning raises questions about whether active selection and support for VE may actually represent the same underlying construct. I suggest conducting a factor analysis that includes all relevant items, encompassing both AS and the support for VE, to assess their distinctiveness.

Additionally, the results, which present correlations rather than causality or directionality, appear somewhat self-evident. It is reasonable to expect that individuals supporting violent extremism would actively seek content that aligns with their views online. Therefore, it is crucial to address whether AS and support for VE truly measure different constructs or if they both reflect the same violent extremist attitudes/intentions.

Despite these concerns, the article contributes to the research community by highlighting the importance of examining different types of exposure and distinguishing between AS and support for VE. However, I recommend that the authors address the key concern regarding the potential overlap in measurement constructs before publication. This clarification will enhance the article's impact and ensure its methodological rigor.

6. PLOS authors have the option to publish the peer review history of their article (what does this mean?). If published, this will include your full peer review and any attached files.

Reviewer #1: **Yes: **Anita Amaliyah

Reviewer #2: No

Reviewer #3: No

Reviewer #4: No

---

## [Author Response · Author response to Decision Letter 0]

27 Sep 2023

Responses to Reviewer Comments

We thank all reviewers for their thoughtful comments, which have helped to improve the quality of the manuscript. In addition to the changes detailed below, we have revised the manuscript to improve readability and style.

Reviewer 1

1) Reviewer 1 noted that the sample should be described in more detail. 

We agree that sample descriptions should always be as comprehensive as possible. We, therefore, had included in the original paper information about participants’ age, gender, and ethnicity. We had also directed readers to the Supplementary Material (specifically S1), which is available through the link to the OSF project page (shared before the sample description), where more information about participants’ level of education, work status, religion, citizenship, and immigration status is presented. We believe that the aforementioned information provides a detailed description of the sample.

Reviewer 2

1) Reviewer 2 highlighted that practical implications of the results are not discussed in sufficient detail. They asked us to address results in the context of UK PCVE efforts.

We agree that it is crucial to discuss practical implications of the findings. Although the data were collected in the UK, we aimed to provide broader points of discussion and, thus, did not focus on the UK explicitly. We added the following two paragraphs to explore implications for risk assessment and PCVE efforts that are implemented online, specifically redirection projects.

“Any practical implications that are based on one study or analysis can only be preliminary. We nevertheless believe that the results inform two fields of practice. First, we contribute insights to the development of individual risk assessment tools that aim to identify those vulnerable to violent extremism before they commit violence (Pressman & Davis, 2022). These tools (e.g., VERA(2) (Pressman & Flockton, 2012); TRAP-18 (Meloy et al., 2021)) typically include a wide variety of risk or protective factors, such as grievances and ideological commitment, criminal history, mental health, or the presence of a social support system. Previous research has further developed approaches to assess individuals’ risk based on textual digital trace data (i.e., linguistic risk assessment; Ebner et al., 2012; Cohen et al., 2014). Our findings suggest that information use behavior, which is to some extent observable, should be considered as well, and in a more nuanced way, in individual risk assessment and management. Namely, we showed that evidence of patterns of only incidental exposure to radicalizing material might be less of a concern, especially if it remains temporary (i.e., individuals are not likely to be embedded in environments where radicalizing messages are regularly disseminated). In turn, information use behavior defined by even a low probability of active selection of radicalizing content online or offline likely indicates stronger support for violent extremism which could manifest in acts of terrorism. Here, efforts to prevent radicalization are especially pertinent.

Perhaps somewhat ironically, in online settings, the fact that individuals actively seek radicalizing information could provide an avenue for targeted prevention. Notably, redirect(ion) projects, which have been implemented in collaboration with large social media platforms, use the search of keywords that are thought to be indicative of (violent) extremist or terrorist actors and narratives as a cue to provide counter-narratives or offer support by civil society organizations specialized in disengagement (Saltman et al., 2021). Evaluations of past programs highlight that their implementation is not without challenges (Helmus & Klein, 2018; Moonshot CVE, 2020). However, our results indicate that using the active search for radicalizing content as one of several indicators of risk is highly promising.” p. 21-22

2) Reviewer 2 also remarked that it would be helpful to introduce other theoretical frameworks (rather than the uses and gratification approach) to provide a rationale for the active selection of belief-congruent information. 

We thank the reviewer for pointing us to those theories that were not yet familiar with. We removed the reference to uses and gratification theories and added a concise rationale grounded in the psychological immune system approach and Meaning Maintenance Model.

“In addition to being incidentally exposed, individuals also actively seek out content, sources, and media. Substantial evidence documents a general preference for information that aligns with existing beliefs and convictions (dissonance theory; Festinger, 1962). One reason for the active selection of belief-congruent information is the defense of one’s sense of self (Porot & Mandelbaum, 2021) and avoidance of threats to one’s self-image (psychological immune system; Gilbert, et al., 1998). The Meaning Maintenance Model (Heine et al., 2006) further stipulates that individuals “are inexhaustible meaning makers” (p. 91) and strive to keep previously established meaning frameworks intact; actively selecting information that confirms one’s beliefs might serve as a strategy to achieve this.” (p. 6)

3) Additionally, Reviewer 2 pointed out that the results for the additional measures of support for violent extremism were not available in the supplementary material.

We believe that this might have been a misunderstanding. The file “S2 Between-group Comparisons for Additional Measures for Violent Extremism.docx” in the folder ‘Supplementary Material’ on the OSF project page includes precisely the information that the reviewer had hoped to find (see https://osf.io/zb2ux).

4) Further, Reviewer 2 asked us to provide more information about the platform that we used for data collection, especially what data quality can be expected.

Responding to this comment, we added a reference to a recent paper that described the use of platforms such as Prolific for the study of violent extremism. Additionally, we added a sentence highlighting that we chose Prolific because it is expected to provide high-quality data.

“Data were collected in September 2021 on Prolific Academic, an online opt-in access panel (Clemmow et al., 2023b). We choose Prolific as a platform for recruiting participants because it has been consistently found to provide high(er)-quality data, especially as compared to other providers such as Amazon Mechanical Turk and CloudResearch (Douglas et al., 2023; Eyal et al., 2022).” (p. 10-11)

5) Lastly, Reviewer 2 made us aware of a typo on page 21, which we corrected. 

Reviewer 3

1) The reviewer pointed out that no references were included in the abstract, and they suggested useful works to be added.

References were not included in the abstract to comply with the journal formatting requirements. On page 3, where the same argument is made again, we do provide several references (Bouhana & Wikström, 2010; Clemmow et al., 2022; Clemmow et al., 2023a; Gill et al., 2015; Hassan et al., 2018; Pauwels & Hardyns, 2018; Pauwels & Heylen, 2020; Pauwels & Schils, 2016; Perry et al., 2018; Rushchenko, 2019; Schils & Pauwels, 2016; Trujillo et al., 2009). We added to this list on page 3 the two recommended works.

2) Reviewer 3 further noted that the definition of radicalization that we provided is too narrow as it suggests that the latter is a natural path towards violence.

We adjusted the wording of the definition and included the recommended book as a reference. 

“Radicalization refers to a process during which individuals adopt extremist attitudes and, in some but not all instances, “come to perceive acts of terrorism as a possible alternative for action” (Wikström & Bouhana, 2016, p. 176; see also Orofino & Allcorn, 2023).” p. 2

3) Further, reviewer 3 highlighted that it would be useful to replicate the study while focusing on specific ideologies.

We agree with this observation, although it is not something that can be achieved in the present research. We extended the call for replications in the discussion by adding a note on ‘different ideologies’:

“Going forward, the patterns that we have identified should be replicated across specific types of radicalizing content and ideologies.” p. 23 

4) Reviewer 3 also reminded us that the study should be ideally replicated in a sample that is known to have adopted extremist beliefs to such a degree that it warrants support by a de-radicalization program. 

We fully agree with this recommendation and extended the suggestions for future research in the following way: “Lastly, it is recommended to explore the research questions with other samples, notably, populations that have adopted (violent) extremist beliefs more strongly than the general population (e.g., individuals who are part of a de-radicalization program) or individuals who have committed or prepared acts of terrorism. In those instances, we would expect the absence of information use patterns that are defined by a near zero probability of incidental exposure to and active selection of radicalizing information or only evidence of incidental exposure.” p. 23

5) Lastly, Reviewer 3 highlighted that the findings might suggest that a higher likelihood of active selection of radicalizing information predicts the dissemination of the latter in other (online and offline) contexts.

We believe that this conclusion is not fully supported by the data, and we are, therefore, hesitant to make it (or add it to the manuscript). We do not have data on what participants do with the information that they actively select; future research could, of course, add such questions.

Reviewer 4

1) Reviewer 4 noted that it is crucial to document that the concepts of IS, AS, as well as the dependent variables are distinct. They recommended that we conduct a factor analysis to confirm that the measured concepts do not ‘overlap’.

We agree with the reviewer and thank them for raising the point. We report an exploratory factor analysis with oblimin rotation in the Supplementary Material (S5) that documents that the scales capturing active selection and incidental exposure assess distinct concepts and that both are distinct from the concept examined with the outcome measures – support for violent extremism. The exploratory factor analysis also shows that all items of the radicalism intentions sub-scale and the violent extremist attitudes scale loaded on one factor, which explains why results for both measures are highly similar. We added a note indicating this point in the manuscript (with a reference to Supplementary Material S5). 

“The highly similar result pattern for both outcome variables is not surprising; indeed, an exploratory factor analysis with oblimin rotation showed that all items measuring the two dependent measures loaded on one factor (Supplementary Material S5).” p. 18

---

## [Editor Report · Decision Letter 1]

20 Oct 2023

Distinct patterns of incidental exposure to and active selection of radicalizing information indicate varying levels of support for violent extremism

PONE-D-23-12940R1

Dear Dr. Schumann,

Your revised manuscript entitled "Distinct Patterns of Incidental Exposure to and Active Selection of Radicalizing Information Indicate Varying Levels of Support for Violent Extremism", which you submitted to PLOS One, has been reviewed.

I have been assigned to be the Guest Academic Editor for this manuscript and am pleased to inform you that following the revisions you have made in response to reviewer comments, your manuscript has been judged scientifically suitable for publication and will be formally accepted for publication once it meets all outstanding technical requirements. Congratulations! Thank you for submitting your work for consideration. 

The comments provided by other reviewers, as well as my own review (to preserve transparency I inform you that I was Reviewer 2) and critical assessment, demonstrate that this work is worthy of publication for several reasons. First, your paper contributes meaningfully to understanding on a critical topic in the understanding of violent extremism: the role of exposure to and pursuit of radicalizing materials in the online context. Second, the methods and analyses you have used are appropriate for the hypotheses, statistically rigorous, and well documented in the main and supplementary materials. Finally, your conclusions are presented in an appropriate fashion, are supported by the data, and are contextualized within the broader literature in a way that will be of use to violent extremism researchers and PCVE practitioners.

Thank you again for submitting this manuscript!

Kind regards,

Daniel W. Snook, Ph.D.

Assistant Professor of Experimental Psychology

Department of Psychology

Florida Gulf Coast University

Guest Editor

PLOS ONE
---

## [Editor Report · Acceptance letter]

25 Oct 2023

PONE-D-23-12940R1 

Distinct Patterns of Incidental Exposure to and Active Selection of Radicalizing Information Indicate Varying Levels of Support for Violent Extremism 

Dear Dr. Schumann:

I'm pleased to inform you that your manuscript has been deemed suitable for publication in PLOS ONE. Congratulations! Your manuscript is now with our production department. 

Kind regards, 

on behalf of

Dr. Daniel W. Snook 

Guest Editor

PLOS ONE